# Transcription, Processing, and Decay of Mitochondrial RNA in Health and Disease

**DOI:** 10.3390/ijms20092221

**Published:** 2019-05-06

**Authors:** Arianna Barchiesi, Carlo Vascotto

**Affiliations:** 1Department of Medicine, University of Udine, 33100 Udine, Italy; barchiesi.arianna@gmail.com; 2Centre of New Technologies, University of Warsaw, 02-097 Warsaw, Poland

**Keywords:** mitochondria, RNA transcription, RNA processing, RNA degradation, mitochondrial diseases

## Abstract

Although the large majority of mitochondrial proteins are nuclear encoded, for their correct functioning mitochondria require the expression of 13 proteins, two rRNA, and 22 tRNA codified by mitochondrial DNA (mtDNA). Once transcribed, mitochondrial RNA (mtRNA) is processed, mito-ribosomes are assembled, and mtDNA-encoded proteins belonging to the respiratory chain are synthesized. These processes require the coordinated spatio-temporal action of several enzymes, and many different factors are involved in the regulation and control of protein synthesis and in the stability and turnover of mitochondrial RNA. In this review, we describe the essential steps of mitochondrial RNA synthesis, maturation, and degradation, the factors controlling these processes, and how the alteration of these processes is associated with human pathologies.

## 1. The Mitochondrial DNA

Given its endosymbiotic bacterial origins, it is not surprising that the organization of DNA in mitochondria is similar to that of bacterial DNA. The bacterial genome is compacted by a factor of 10^4^-fold that of its volume to form the bacterial nucleoid, and in a similar way the mitochondrial DNA (mtDNA) is compacted and organized in discrete protein–DNA complexes distributed throughout the mitochondrial matrix [1]. mtDNA was first described in the 1960s [2] and completely sequenced in 1981 by Anderson et al. [3]. Despite the differences between nucleoids in mammals and in yeast, most of the information on the structure and composition of the nucleoid comes from studies on yeast that have been fundamental in understanding the mammalian nucleoids. During the 1970s, scientists were able to stain the mtDNA in vivo and noticed sub-mitochondrial structures near the inner membrane, later called nucleoids, where the mtDNA was located. Each nucleoid can contain more than one mtDNA molecule and each mitochondrion can contain tens (for yeast) or hundreds (for mammalians) of nucleoids. Moreover, each nucleoid can be considered as a unit of genetic segregation in the mitochondria, as every nucleoid acts as an independent genetic unit.

Mitochondrial DNA comprises 0.1–2% of the total DNA in most mammalian cells. There are several unique features of the mtDNA: human mtDNA is circular, 16 kbp long, and inherited from the mother. It encodes two rRNAs, 22 tRNAs, and 13 proteins, all of which are involved in the oxidative phosphorylation process [4]. The intragenic sequence is almost absent or limited to a few bases [5] and mtDNA does not have histones, instead it is organized in nucleoid structures. A large number of experiments showed that multiple copies of mtDNA could be found in each nucleoid, usually from two to 10 copies each, depending on the cell line studied [6]. However, quantitative analysis of the size and mtDNA content of the nucleoid in cultured mammalian cells suggests that an average nucleoid may contain five to seven mtDNA molecules packed in a space of 70 nm [6], similar to the bacterial genome. Tight packaging of the mtDNA is achieved thanks to the proteins present in the nucleoid, such as the mitochondrial transcription factor TFAM.

## 2. The Transcription Process

Two different strands can be recognized in the mtDNA: the heavy strand rich in guanine bases, which also contains the majority of mitochondrial coding genes, and the light strand, encoding only for the MT-ND6 (NADH-ubiquinone oxidoreductase chain 6) protein and eight tRNAs. Both strands are transcribed at the same time, giving origin to very long transcripts, of almost mtDNA length, that are subsequently processed. Transcription seems to take place in the nucleoids due to the presence of the mitochondrial transcription machinery. However, experiments performed with 5-bromouridine (BrU) aimed at tracking the progress of the nascent RNA transcripts showed that newly transcribed mtRNAs are also found in discrete foci situated in close proximity to mitochondrial nucleoids, called mitochondrial RNA granules [6,7,8], which will be described in detail later. 

The essential enzymatic machinery dedicated to the mtDNA transcription is completely different from the nuclear one and is composed by few proteins: the DNA-directed RNA polymerase (POLMRT) [9,10], the mitochondrial transcription factor A (TFAM) [11,12], the mitochondrial transcription factors B1 and B2 (TFB1M and TFB2M) [13,14], the mitochondrial transcription elongator factor (TEFM), a single strand binding protein (SSBP), and the family of mitochondrial transcription termination factors (mTERF1-4). Additionally, MRPL12 (39S ribosomal protein L12) acting as a complex with POLMRT and TEFM is required for the processivity during the elongation phase and to prevent the termination of the transcription at conserved sequence block 2 (CSB2) [12,15,16]. It is in the non-coding region of the D-loop that mitochondrial translation is regulated by the heavy strand promoters 1 and 2 (HSP1 and HSP2), the light strand promoter (LSP) [17], the transcription termination-associated sequences TAS1 and TAS2, and the conserved sequence blocks1 and 2 (CSB1 and CSB2). HSP2 is the sequence promoting the transcription of almost full-length of the heavy strand [18,19] while HSP1 is only responsible for the transcription of the two ribosomal RNAs that terminate just after the tRNA^Leu^. Regarding the transcription of the light strand, LSP1 is a unique site where the transcription can start and terminates at CSB1, but the majority of initiation events from this site stop 200 bp downstream the promoter, at the CSB2 site [20]. CSB2 is a conserved G-rich sequence and its function is to terminate the transcription of the 7S RNA for the replication of the mtDNA, regulating the switch between mtDNA replication and transcription [20,21]. 

Transcription is initiated by the binding of TFAM to an high affinity site −10/−15 bp from the site of transcription start, creating a stable U-turn [22]. Once in this position, the POLMRT can bind directly to the TFAM unit that recruits the protein to the promoter. POLMRT can slide in the DNA from positions −50 to −60, approximately, and the complex is unable to start the process until TFB2M binds the complex TFAM–POLMRT and the fully assembled initiation complex encircles the promoter. In fact, TFB2M is required for the conformational changes needed for the transcription starting [23]. Among all genes involved in mitochondrial translation, TFAM is the only one whose mutation has been shown to cause human diseases. Mutations in the TFAM gene are linked to an autosomal recessive disorder with infantile-onset progressive liver failure. mtDNA copy number is decreased, and patients present defects in the respiratory processes [24]. Mutation of other components of the transcription machinery seems not to be causative of any pathology in humans. TFAM is also involved in neurodegenerative pathologies, controlling the mtDNA copy number [25].

To enhance the processivity of the translation process, TEFM can interact with the catalytic, C-terminal part of POLRMT, and depletion of TEFM impairs transcription elongation both in vitro and in knock-down cell lines [16]. TEFM stimulates POLRMT interactions with an elongation-like DNA:RNA template, and the protein is present at the promoter before the initiation of transcription. It also helps the POLMRT with the transcription of long stretches of RNA and to bypass regions with secondary RNA structures (e.g., tRNAs) [15,26]. It was recently demonstrated that TEFM is able to enhance mtRNAP transcription elongation by reducing the frequency of long-lived pauses and shortening their durations and helps to overpass the CSB2 sequence to continue the transcription of the full polycistronic RNA [26].

The process of mitochondrial transcription termination is still unclear. There is still a debate if MTERF1 is really needed for the termination of all the transcription processes that originate from the three different promoters of the control region. Recent studies have shown that knock-down mice for the *MTERF1* gene do not have any notable consequence on the phenotype, and the levels of ribosomal and messenger RNA are unaffected. However, biochemical studies have shown that MTERF1 only partially terminates H-strand transcription [27], whereas transcription in the opposite direction (L-strand transcription) is almost completely blocked (Figure 1).

## 3. Regulation of Transcription by Protein Direct Binding to mtDNA

Many different proteins are involved in the regulation of transcription, such as hormones, nuclear transcription factors, and chromatin remodeling enzymes which are also able to interact with the mitochondrial DNA, and RNA/DNA modifying enzymes. Here we propose a brief overview of the mtDNA transcription regulation operated by these factors (Table 1).

### 3.1. Hormones

One of the first proteins discovered to be involved in the regulation of transcription is the thyroid hormone T3, which is able to promote the mtDNA transcription by directly binding the mtDNA genes [28]. More recent is the observation that a dominant negative form of the thyroid hormone is able to bind the thyroid receptor elements present in the 12S gene and in the D-loop region [29,30,31]. The cAMP-responsive element binding protein (CREB) was found to be able to specifically bind to a natural or a palindromic thyroid-responsive element. Moreover, this protein specifically binds to a direct repeat 2 sequence located in the D-loop of the mitochondrial genome [32]. Glucocorticoid hormones were also found to be in mitochondria where they modulate the transcription binding to the glucocorticoid receptor present in the mitochondrial inner membrane [33,34]. The estrogen receptor (ER) was found in the mitochondria of cardiac cells. It was hypothesized that E2 (17β-estradiol) and ERβ-mediated cardioprotection was dependent on mtDNA transcription encoding for mitochondrial respiration activity. It was also demonstrated that E2 can also increase the ER β mtDNA binding activity followed by an increase in complex V encoding gene expression [35]. Melatonin was also recently described as a potential hormone that can control the mtDNA expression through the reduction of several mitochondrial transcription factors. It was demonstrated that melatonin was able to decrease, at both mRNA and protein levels, TFAM expression as well as other proteins such as transcription factors TFB1M and TFB2M, interfering with mtDNA transcription [36]. 

### 3.2. Nuclear Transcription Factors

Several nuclear transcription factors are also able to bind the mtDNA. Recently, Marinov and colleagues through chromatin immunoprecipitation (ChIP) sequencing from seven different human cell lines demonstrated the presence of diverse transcription factors to be bound to the mtDNA. From this study, the most enriched ones were CEBPb (CCAAT/enhancer-binding protein beta), c-Jun, JunD, MafF, MafK, Max, NFE2, and Rfx5 (DNA-binding protein RFX5). Most of them were found to be enriched in the D-loop structure as expected, but some were also found to be able to bind sequences in proximity of the OXPHOS (Oxidative phosphorylation) subunits encoding genes with differences between the cell lines studied [37]. Some of them have been better characterized for their function in regulating the mitochondrial transcription. c-Jun, a well-known nuclear transcription factor, was shown to decrease mtDNA transcription in concert with other enzymes involving the retinoid X receptor (RXR) pathway [38]. NFATc1 (Nuclear factor of activated T-cells, cytoplasmic 1) is a transcription factor involved in the differentiation of human mesenchymal stem cells in osteoblasts. This protein was also found to be in mitochondria and to interact with the D-loop region inhibiting the transcription of some crucial genes such as *Cyt-b* (Cytochrome b) and *MT-ND1* (NADH-ubiquinone oxidoreductase chain 1), therefore acting as a negative regulator of mtDNA transcription during the calcification process [39].

### 3.3. Chromatin Remodeling Enzymes

It is well established that, in eukaryotes, methylation on cytosine at the CpG sequences in the nuclear DNA (5mCpG) regulates the transcription of the genes through alteration of the chromatin structure. However, mtDNA is devoid of histones and nucleosomal chromatin, so the mechanisms by which these CpG islands are methylated in mitochondria must be different, but they are still unknown. As mentioned before, the best-characterized and abundant protein binding duplex DNA is the HMG-box protein TFAM, which is permanently associated with mtDNA [40]. TFAM is the only factor that plays a clear structural role in mtDNA organization in nucleoids, similar to the role of histones for the nuclear DNA or the histone-like proteins in bacteria [41,42]. This protein is probably the major factor responsible for the tight packaging of the mtDNA, so it plays a role in mtDNA topology [43]. TFAM is able to bind the double helix of the DNA and acts as a packaging protein mostly on the central region of the mtDNA because the estimation of TFAM concentration is too low for the coverage of the entire mtDNA molecule [41,44,45]. It was estimated that one molecule of TFAM can bind the DNA in regular intervals of 20 bp [46,47,48] but since the protein acts as a homodimer, two molecules of TFAM bind the DNA in intervals of 35–40 bp [46,47]. Although it has been suggested that TFAM lacks binding sequence specificity, some researchers proposed mtDNA binding site preferences [49], especially at regions that tend to adopt G-quadruplex structures (GQP) in vitro [50] but not in vivo [51].

Due to the tight coverage of the mtDNA with TFAM and to the non-sequence specificity of the interaction, this protein must interact with many CpG islands in the mtDNA. There is a growing amount of evidence showing that some of these CpG islands could be methylated and that this can influence the transcription of genes possessing this modification, similar to what happens in the nucleus. It was proposed by Minczuk and colleagues that the occurrence of 5mCpG has the potential to impact TFAM-mtDNA recognition in mammalian cells. The authors demonstrated that the methylation of CpG sequences in the HSP can increase the binding activity of TFAM inducing TFAM multimerization at these sites, without changing the compaction of the DNA. Additionally, 5mCpG seems to have a clear and context-dependent effect on transcription: it was shown that CpG methylation in the HSP1 promoter strongly increases the transcription starting from this site [52]. It was recently suggested that mtDNA chromatin-like organization is gradually established during the embryogenesis of mammalian cells, showing an increasing accumulation in the density of footprinting sites, revealing a dynamic achievement of mtDNA coverage during embryonic development and in the regulatory regions (D-loop) [53]. In fact, not only the methylation of CpG sequences but also different chromatin remodeling enzymes can be found in the mtDNA. 

MOF (males absent on the first; also known as MYST1 or KAT8) is the major lysine acetyl transferase (KAT) responsible for the deposition of H4K16ac in flies and mammals [54,55,56], and it was shown to directly bind mtDNA controlling mitochondrial transcription. In its absence, mitochondrial transcription is downregulated with catastrophic consequences on metabolism and respiratory function [57]. On the contrary, STAT3 (signal transducer and activator of transcription protein 3) is able to bind the mtDNA and TFAM, which influences the transcription of mitochondrial genes. It was demonstrated that the ablation of STAT3 in keratinocytes results in an increased mitochondrially encoded gene transcripts [58]. SIRT1 (NAD-dependent protein deacetylase sirtuin-1) was also found to be present in mitochondria-forming stable complexes with TFAM inside the nucleoids and to be able to bind the mtDNA [59]. 

Dnmt (DNA methyl-transferase 1) is a methyl transferase enzyme commonly associated with gene silencing and was found to accumulate in the mitochondria of retinal endothelial cells treated with high glucose concentrations. In the same background, methylation of the mtDNA was also increased in the D-loop causing a decreased transcription of the mtDNA, resulting in dysfunctional mitochondria and accelerated apoptosis [60].

### 3.4. MitomiRs

MicroRNAs (miRNAs) are a class of small RNAs able to regulate gene expression through interference with the translation process. A particular class of these miRNAs (mitomiRs) were found to be enriched in proximity of mitochondria and inside these organelles, controlling the mitochondrial behavior [61]. It is still not clear how mitomiRs are imported into the organelle, but it seems that the polynucleotide phosphorylase (PNPase), a ribonuclease present both in the matrix and in the intermembrane space of the mitochondria, is the protein deputized for this role [62]. Several studies have identified the mtDNA as a source and target of mitomiRs [63,64,65]. Although the canonical mechanism by which these molecules exert their function is the prevention of translation, it was proposed that they can also act directly on the mtDNA to block the transcription process. It was in fact demonstrated that mitomiR 2392 can regulate the gene expression of mtDNA in tongue squamous carcinoma cells. The presence of this miRNA in mitochondria is able to reverse the chemoresistance of these cells, directly binding, with Argonaut 2, to specific sequences in the H-strand of the mtDNA, partially blocking the transcription, resulting in the reduction of OXPHOS complexes activity [66].

## 4. Nuclear Factors Indirectly Influencing Transcription

Many factors are involved in the mitochondrial transcription regulation and the crosstalk with the nucleus. Nuclear proteins play a central role in the indirect control of the mtDNA transcription and mitochondrial function. The most studied factors influencing mitochondrial activity are the nuclear respiratory factors 1 and 2 (NRF1 and NRF2) and yin yang 1 (YY1), along with p53 and the peroxisome-proliferator-activated receptor coactivator (PGC1).

NRF1 and NRF2 have been linked with the transcription of many genes involved in mitochondrial function, not only the nuclear encoded subunits of the respiratory chain but also proteins belonging to the transcription machinery [67]. It has been demonstrated that they are able to enhance the transcription of mitochondrial biogenesis genes during exercise in rats and in response to diverse stress challenges [68,69]. TFAM and mitochondrial RNA processing enzymes are target genes of NRF1, and TFBM1 and TFBM2 were also recently demonstrated to be their target. These two proteins enhance mtDNA transcription in the presence of TFAM and mitochondrial RNA polymerase. Moreover, it has been shown that the same NRF-recognizing sites are also needed for the activation of PGC1 enzymes and PRC (PGC-1-related coactivator) activation and that TFB genes, TFAM and PCG1 or PRC, are upregulated in cells where mitochondrial biogenesis is induced [70]. 

The physiological activation of these PGC1 family members and PRC coactivators is also required for inducing and integrating signals controlling cell growth, metabolism and cell differentiation, and mitochondrial biogenesis [71]. One of the targets of PGC1α is YY1, which in concert with the mammalian target of rapamycin (mTOR) can regulate the mitochondrial transcription. mTOR is a kinase and is a fundamental component in nutrient sensing and energy pathways in the cell. mTOR also contributes to the control of mitochondrial oxidative activities [72]. Inhibition of the mTOR protein in skeletal muscle cells results in the loss of PGC-1a expression and nuclear respiratory factors, leading to a decrease in mitochondrial genes expression and OXPHOS impairment. This function is mediated by YY1, which was demonstrated to be required for rapamycin-dependent repression of respiration genes, a common target of mTOR and PGC1α [73]. 

The tumor suppressor p53 has a well-known role in the maintenance of the mitochondrial DNA, contrasting mtDNA mutagenesis [74]. p53 was also demonstrated to be involved in controlling the mitochondrial mtDNA copy number. In fact, p53 was firstly described to be in mitochondria in 2012 with a role in the start of the necrosis process [75]. The loss of p53 in null mouse and knock-down human fibroblasts determines the mtDNA depletion and reduction of TFAM, compromising transcription, protein synthesis, respiration, and mitochondrial mass [76]. Moreover, it was shown that p53 is able to inhibit the entrance of the NF-κB (Nuclear factor kappa-light-chain-enhancer of activated B cells) family member RelA into the mitochondria, impeding the repression of mitochondrial transcription actuated by this protein in the absence of p53 [77].

Another key factor in the regulation of mitochondrial function is the HIF1α (hypoxia inducible factor-1α) protein. This protein usually mediates the adaptation of the cell to conditions such as hypoxia and oxidative stress through regulation of gene expression. The way by which HIF1α regulates cell responses to these stresses in mitochondria is still not well understood. It was shown that HIF1α was able to translocate inside mitochondria after induced oxidative stress in human cells. Moreover, overexpression of a mitochondrially targeted form of the protein helps to attenuate apoptosis and promote expression of mtDNA-encoded mRNAs independently from the expression of nuclear OXPHOS subunits [78].

Finally, a higher mechanism to control mitochondrial gene expression is the co-expression of mitochondrial genes and the nuclear genes encoding for the respiratory chain complex proteins. Both the genetic systems can adapt to enhance or reduce the transcription depending on the energy demand of the cell [67,79,80]. It was demonstrated that there is a co-expression of genes inside each one of the five OXPHOS complexes from both the nuclear and mitochondrial parts. Although common sequences (core promoters) were found in the promoter regions of the OXPHOS genes, the authors were unable to find a common pattern of OXPHOS-specific transcription factors for each set of genes expressed [81]. 

## 5. Processing of Mitochondrial Transcripts

From their synthesis to their degradation, mtRNAs undergo several stages of maturation and modification for the correct production of mtDNA-encoded proteins. As mitochondrial DNA replication and transcription need to be spatio-temporally regulated to adapt to the metabolic demand of the cell, so must the basic stages of mitochondrial gene expression. To achieve this, mitochondria restrict mtRNA processing and maturation to dynamic protein structures called mitochondrial RNA granules (MRGs), which provide a regulatory function for post-transcriptional processing, allowing all mtRNAs to be fully mature before protein synthesis [82,83]. To confirm and better understand the function of these structures, in 2015 Antonicka et al. [84] characterized the proteome of the granules using GRSF1, a core component of the granule [7,8], as a bait Some proteins found by the authors were already confirmed in the literature, such as RNaseP [8] and more interestingly the so-called “mitochondrial degradosome” composed of hSUV3 (ATP-dependent RNA helicase SUPV3L1) and PNPase [85], confirming the hypothesis that MRGs are not only sites of RNA processing, but also of RNA degradation and turnover. Mass spectrometry analysis of immune-precipitated fractions showed a large number of proteins responsible for the post-transcriptional processing of the primary polycistronic transcript, such as MRPP (Mitochondrial ribonuclease P protein) 1, -2, and -3, RNA-modifying enzymes such as TFB1M (Dimethyladenosine transferase 1), PTCD3 (Pentatricopeptide Repeat Domain 3), and the mitochondrial poly-A polymerase. In addition, proteins belonging to the mitochondrial translation machinery, as well as structural proteins of the small (mt-SSU) and large (mt-LSU) mitochondrial ribosomal subunits, aminoacyl tRNA synthetases, and factors involved in ribosome assembly and disassembly, were present in the analysis. These data suggest that MRGs are also involved in mitochondrial ribosome biogenesis and in mitochondrial translation regulation, with a function analogous to that of the nucleolus, where initial steps of ribosomal assembly are performed [86,87]. It is possible that both mtDNA and its transcription products are portioned within non-membrane bound compartments to provide a greater degree of spatio-temporal regulation of mtRNA processing. The last stage of mtRNA life was suggested to take place in specific foci, called D-foci (degradation foci), composed mostly of the mitochondrial degradosome [85,88]. It has been shown that these foci co-localize with the MRGs, although it is still not clear whether they form a subset of MRGs or are separate entities with a distinct composition and purpose. In these structures, several catalytic mitochondrial enzymes and other mitochondrial and non-mitochondrial proteins, whose role remains to be established, were also found. 

The transcription of mtDNA gives rise to two polycistronic transcripts that must be somehow processed to release different RNA species. Most of the mRNAs and mt-rRNA-coding regions are separated by mt-tRNAs. These RNAs are separated from each other according to the generally accepted mt-tRNA punctuation model [3,5]. The mitochondrial RNA-processing machinery initiates the cleavage of the mt-tRNA sequences, freeing the mt-rRNAs or mt-mRNAs that they intersperse. However, not all the mRNA and are flanked by a mt-tRNA coding sequence, such as ATP6/8 and COIII or ND5 and Cyt B.

It has been recently proposed that the early stages of the mitochondrial transcription can take place co-transcriptionally inside the mitochondrial RNA granules. Indeed, most of the proteins involved in mtRNA processing were found to be part of the MRG proteome [7,8,89]. In particular the 5′-end of the mt-tRNAs is processed by the protein complex RNaseP, composed of MRPP1, -2, and -3 that are found in the granules [8,90]. MRPP1 is a m^1^G9-methylase, while MRPP2 is a dehydrogenase also involved in other cellular functions [91]. These two proteins form a subcomplex that also participates in tRNA modification [92]. MRPP3 is responsible for the hydrolysis of the phosphodiester bond [93]. The knockout of any of these proteins causes an accumulation of the RNA precursor molecule, reducing the steady-state levels of the mature form of mt-tRNAs and some mt-mRNAs [94]. Mutations of the gene encoding for MRPP1 (TRMT10C) were reported in infants presenting at birth with lactic acidosis, hypotonia, feeding difficulties, and deafness. Fibroblasts from individuals carrying missense mutations of the gene show decreased levels of mtRNA precursors, indicating an impaired mtRNA processing and an inefficient mitochondrial protein synthesis [95,96]. In addition, a novel mutation in the MRPP2 encoding gene (X-linked gene, HSD17B10) was found to be causative of intractable epilepsy and global developmental delay. The pathogenicity of the mutation is due to a general mitochondrial dysfunction caused by the reduction in maturation of mt-tRNAs [95,97].

RNaseZ (also named Zinc phosphodiesterase ELAC protein 2 (ELAC2)) is responsible for the 3′-end processing of the mt-tRNAs, but this protein was not found to be part of the MRG proteome [98,99]. Mutations in the ELAC2 gene were confirmed to be the cause of an infantile hypertrophic cardiomyopathy characterized by complex I deficiency and accumulation of mt-tRNA precursors in skeletal muscle and fibroblasts as well as impaired mitochondrial translation [100]. It was then proposed that the primary transcripts undergo an initial round of processing, partially co-transcriptionally, inside the MRGs, while a second part of the maturation takes place later and outside the granules. In this case, the knockout of the protein causes the accumulation of mtRNA precursors [94,99].

Recently, other proteins were described in the MRGs that could have a function in the processing of the primary transcript. GRSF1 (G-rich sequence factor 1) is an RNA-binding protein that has been shown to co-localize with newly synthesized mtRNA and with MRPP1. The loss of the protein results in a decrease in the mature form of some transcripts [7,8]. PTCD1 is another protein implicated in the mtRNA metabolism. This protein directly interacts with RNaseZ and seems to play a role in the coordination of 3′-end processing [94,101].

mt-tRNAs released from the primary transcript need to reach their mature form and a stabilized conformation to be used during the translation process. mt-tRNAs undergo a different variety of chemical modifications that can be divided into two categories: those able to confer the tRNA the correct structural stability and folding and those that coordinate the proper tRNA function altering their interaction with other factors [102]. The mammalian mitochondrial ribosome is composed, like the cytosolic one, of two subunits of different sizes, the small subunit 28S (mtSSU) and the large subunit 39S (mtLSU) [103,104,105,106]. Both of them are composed of the mitochondrially encoded rRNAs 12S and 16S, respectively, and the ribosomal proteins that are codified by nuclear genes and imported into the mitochondrial matrix. Mitochondrial rRNAs undergo post-transcriptional modifications to be functional; however, differently from the cytosolic ones, the range of modifications is less wide, and they do not require any nucleolytic processing [102,107,108] (Figure 2). 

## 6. Maturation of Mitochondrial mRNA

Once released from the primary transcript, mt-mRNAs undergo post-transcriptional modification. Stabilization of the mitochondrial mRNAs is very different and simpler compared with the process of nuclear-encoded mRNAs. On the contrary, the degradation process shows similarities to the compartmentalization and the degradation process that takes place in the cytoplasm [102].

### 6.1. Polyadenylation

The first difference between the maturation process of the nuclear-encoded mRNAs and that of the mitochondrial ones is the lack of 5′CAP modification as well as the absence of introns. The 3′-end of mt-mRNAs is modified to have a poly-A tail much shorter than nuclear mRNAs that can go from 45 to 55 nucleotides, with some exceptions [109]. Indeed it was found that ND6 completely lacks the poly-A tail and is immediately competent for translation, while ND5 can be either oligo-adenylated or not adenylated at all [109]. Poly-A tails are synthesized by a non-canonical poly-A polymerase (mtPAP) that localizes in MRGs, suggesting again that the first maturation steps take place co-transcriptionally in the granules [110,111,112,113]. The knock-down of mtPAP or a decrease in the polyadenylation leads to impaired mitochondrial translation and the disruption of the mitochondrial respiratory function [111]. The exact role of the poly-A tail in mitochondrial transcripts is still unclear [114]. Seven out of 13 mt-mRNAs do not encode a complete stop codon for translation termination. Most of them are cut out when the mt-tRNAs are nucleolitically cleaved from the primary transcript, often leaving a “U” or “UA”. For this reason, it was suggested that the poly-A tail added after that cleavage has the function to complete the stop codons [3,5]. Although the classical role of the poly-A tail is to stabilize and increase the half-life of the transcripts; in the mitochondrial compartment, this is not always the case. It was shown that the poly-A tail can decrease the stability of Cytochrome c oxidase subunit I, II, III (COI, COII, COIII), and ATP6/8 (ATP synthase protein) and increase the stability of ND1, ND2, ND3, ND4, ND4L, ND5, and Cyt. B. The mechanism of this transcript-specific role of polyadenylation remains to be elucidated [115,116,117]. A mutation in the gene encoding mtPAP was found to cause a form of spastic ataxia with optic atrophy in the Amish population. RNA extracted from the cells of these patients showed shortened polyA tails, which consequently caused an aberrant and inefficient translation and a defective activity of the respiratory chain complexes [118]. OligoA polymerization was retained by the mutated protein, and it was demonstrated that this difference in the polyA tail has differential effects on transcript stability that were dependent on the particular species of transcript [113]. The phosphodiesterase 12 (PDE12) protein, a mitochondrial 2′ and 3′ phosphodiesterase [115,119], is responsible for the removal of the poly-A tail, but this activity was shown only in vitro and in cultured cells, after overexpression of the protein [115]. 

### 6.2. Non-Canonical Capping

In eukaryotic cells, mRNAs are normally capped with a 5′ end N7 methyl guanosine (m7G) cap that promotes their translation and stability. Mitochondrial mRNAs do not have this type of modification. It was demonstrated that in human cells, like in *Escherichia coli* [120], mRNAs can harbor a 5′-end nicotinamide adenine dinucleotide (NAD+) cap [121], which in contrast to the m^7^G cap does not support translation but instead promotes mRNA decay [122]. This kind of modification was recently detected also in mitochondria of human cells. It was also found that POLMRT is able to add an NAD+ cap to mitochondrial RNAs species in vitro and that the enzyme is able to initiate transcription of NADylated RNA from the LSP promoter. This suggests that the NAD+ capping can influence both the translation and the replication process, given that the majority of transcription events that originate from the LSP promoter give rise to the 7S RNA needed for the replication of mtDNA [123].

### 6.3. Processing Regulation by MRG Proteins

The G-rich sequence factor 1 (GRSF1) is one of the first mtRNA granule proteins to be described. This protein preferentially binds RNAs derived from three genes (the ND6 mRNA and the long noncoding RNAs for Cyt. B and ND5, from the LS) and all these genes contain consensus sequences for GRSF1. It was demonstrated that the silencing of this protein causes alterations in mitochondrial RNA stability, aberrant mtRNA processing, abnormal loading of mRNAs and lncRNAs on the mitochondrial ribosome, and impaired ribosome assembly [7]. It was also shown that GRSF1 can interact with RNase P, thus being involved directly into the processing of the primary transcripts [8]. These data implicate GRSF1 as a key regulator of post-transcriptional mitochondrial gene expression. 

The FASTK (FASTK and FAS-activated serine/threonine kinase 1–5) family proteins are non-canonical structured RNA binding proteins, all of which are located in mitochondria. All family members have a different role in the regulation of mtRNA biology, from processing to translation [124]. FASTK is localized both in the nucleus and in mitochondria, where it interacts with GRSF1 and ND6 mRNA. Since ND6 mRNA is one of the non-canonical transcripts, it was proposed that FASTK binds this RNA protecting it from the PNPase-hSUV3 complex-mediated degradation. The knock-down of FASTK results in the loss of ND6 mRNA and in decreased activity of complex I [125]. FASTKD1 interacts with Twinkle and co-localizes with mtDNA [126]. Its loss causes ND3 mRNA accumulation and an increased complex I activity, opposite to the FASTK loss effect. How FASTD1 negatively regulates complex I activity through the effect on ND3 mRNA is still unknown [126,127]. FASTKD2 is present in MRGs, where it interacts with the GRSF1 protein. It binds with a defined set of mitochondrial transcripts including 16S ribosomal RNA and ND6 mRNA [84,125,128]. More recently, FASTKD2 was found to be part of a pseudouridilation functional module inside the mitochondria along with NGRN (Neugrin), WB-SCR16, and the Mitochondrial mRNA pseudouridine synthases RPUSD3, RPUSD4, and TRUB2 [129]. This module is fundamental for the stability of 16S rRNA [129]. The loss of FASTKD2 leads to a decrease in 16S rRNA, an impaired translation process, and the aberrant processing and expression of ND6 mRNA. Depending on the cell line considered, the depletion of FASTKD2 can cause either a decrease in the activity of the respiratory complexes or no visible effect [84,128,129]. 

Of the entire FASTK protein family, FASTKD2 is the only one whose mutations can cause human syndromes. A homozygous nonsense mutation in the FASTKD2 gene was found to cause mitochondrial encephalomyopathy associated with developmental delay, hemiplegia, convulsions, and low cytochrome C oxidase activity in skeletal muscle [124]. Recently, a heterozygous mutation was reported in a case of adult-onset MELAS (mitochondrial encephalomyopathy, lactic acidosis, and stroke-like episode)-like syndrome [130]. FASTKD2 has been also implicated as a target for modulating neurodegeneration and memory loss in ageing and dementia [118]. Furthermore, FASTKD2 has been also shown to mediate apoptosis in breast and prostate cancers. FASTKD2 is the last target of the NRIF3/DD1/DIF-1 pro-apoptotic axis, and this pathway was found to mediate the apoptosis in LNCaP cells.

FASTKD3 is also required for the correct processing of the transcript, and its silencing leads to an increased steady-state level and a half-life of ND2, ND3, CYB, CO2, and ATP8/6 mRNAs. It is also required for COXI protein synthesis and for the proper assembly and activity of the complex IV [131]. FASTKD4 is present overall in the mitochondrial matrix and it binds to the majority of heavy-strand encoded transcripts. Its depletion leads to decreased levels of ATP8/6, CO1, CO2, MT-CO3, ND3, CYB, and ND5 mRNAs regulating the stability of these transcripts [127,132]. FASTKD5 is enriched in MRGs and is required for the processing of some of the non-canonical transcripts. The loss of FASTKD5 results in the accumulation of partially processed primary transcripts such as ATP8/6-CO3, ND5-CYB, and ribosomal RNAs-CO1 and to an overall decrease in protein translation, suggesting a regulation of abundance of all mitochondrially encoded RNAs [84]. FASTKD5 was also proposed as a regulator of adaptation during metabolic stress, oncogenic transformation, and innate immunity through the association with NLRX1, a member of the Nucleotide-binding Leucine-rich Repeat family receptor (NLR) usually implicated in the immunity response as a negative regulator of anti-viral signaling [133]. It was demonstrated that the association of NLRX1 with FASTKD5 has a negative impact on the non-canonical mtRNA transcript processing of mitochondrially encoded proteins belonging to complex I and IV and consequently on the activity of these complexes [133].

### 6.4. Processing Regulation by Nucleoid Proteins

Recently, it was proposed that Twinkle and the mitochondrial Single-stranded DNA-binding protein (mtSSB), two proteins normally associated with nucleoids and mtDNA, could play a role in the mitochondrial RNA biology. Indeed, both of them were found in the MRGs, and repression of either of them was demonstrated to alter mtRNA metabolism. In particular, Twinkle depletion leads to the disruption of MRGs without disturbing RNA expression and processing, while mtSSB loss leads to RNA processing defects, an accumulation of mtRNA intermediate products, and increased levels of dsRNA and RNA/DNA hybrids [134].

## 7. Mitochondrial mRNA Degradation

Once the mRNAs, tRNAs, and rRNAs are used several times for the translation of proteins, they might be degraded to eliminate aberrant or damaged transcripts. The best characterized protein complex dedicated to the mt-mRNA degradation in the mitochondrial matrix is the hSUV3/PNPase complex [85]. hSUV3 and PNPase have been shown to partially co-localize with the MRGs, although it has been suggested that RNA degradation can take place in specialized foci, called D-foci [83,102]. D-foci, besides containing the degradosome, also localize with newly synthetized mtRNA, similarly to MRGs, suggesting that a subpopulation of MRGs can participate in the RNA processing of degradation mediated by the degradosome [85]. Another protein potentially involved in the degradation of mtRNA is the RNA exonuclease REXO2. This 3′-to-5′ exonuclease acts as a homotetramer and degrades oligonucleotides in the matrix. As for PNPase, REXO2 seems to have a dual localization: in the mitochondrial matrix and in the mitochondrial intermembrane space [135]. It was suggested that, because the degradosome is expected to degrade RNA in small oligo-ribonucleotides, it is possible that they become a substrate for REXO2 to complete later stages of decay [135]. 

hSUV3 is an NTP-dependent helicase that has more than one isoform, and at least one of these is localized in the mitochondrial matrix. This protein is able to unwind different DNA and RNA substrates. Szczesny et al. suggested that this helicase is involved in the degradation of the damaged mtRNAs and plays a role in the decay of the properly processed RNA molecules [136,137,138]. The protein partner acting with hSUV3 is PNPase, a polynucleotide phosphorylase capable of 3′-to-5′ phosphorolysis and 5′-to-3′ RNA polymerization [139,140]. For its role in RNA degradation, PNPase localizes in the mitochondrial matrix, but some studies have recently shown the localization of the enzyme in the mitochondrial intermembrane space [141]. For this multiple localization, PNPase has been attributed to different processes in RNA metabolism. In the mitochondrial matrix, it takes part in the degradation process of RNA and in the polyadenilation process, while in the intermembrane space it seems to be involved in the import of different RNA species from the cytoplasm [85,141,142]. PNPase co-immunoprecipitated with SUV3 from mitochondrial cell extracts and foci of exogenously produced hPNPase and SUV3 colocalized with mitochondrial DNA and RNA [85,88]. Furthermore, the knock-down of hPNPase in HeLa cells resulted in the stabilization of mitochondrial mRNAs [85,143], while the depletion of hPNPase or SUV3 led to the accumulation of mitochondrial double-stranded RNA [144]. It was also demonstrated recently that the PNPase/hSUV3 complex is able to interact with GRSF1 to degrade the non-coding antisense RNAs produced with the transcription process [145].

Pathogenic mutations of the PNPase encoding gene were identified and gave rise to two different pathologies. Gln387Arg transition leads to severe hypotonia and movement abnormalities in late infancy, with severe but non-progressive encephalopathy accompanied by elevated plasma and cerebrospinal-fluid lactate levels [146]. The Glu475Gly transition leads to severe, early onset hearing impairment in early childhood [147]. It was demonstrated that these mutations affect the homo-trimerization of PNPase, disturbing the RNA import function of the protein, but the effect on exoribonuclease activity has not yet been investigated.

REXO2 is the other exoribonuclease present in the intermembrane space and in the mitochondrial matrix and has a 3′–5′ exonuclease activity specific to small oligomers. It was shown that silencing REXO2 leads to a disorganized mitochondrial network, a decrease in mtRNA and mtDNA levels, and an impaired translation process, negatively affecting cell growth [135]. 

## 8. Regulation of Mitochondrial RNA Stability and Decay

The regulation of mRNA stability and turnover are fundamental in controlling gene expression and are usually mediated by protein complexes. The best characterized complex is the LRPPRC/SLIRP, which prevents the degradation of mtRNAs. LRPPRC is a leucin-rich pentatricopeptide repeat (PPR)-containing protein that binds RNA and is mainly present in the mitochondrial matrix [148]. The knock-down of LRPPRC in mice results in drastically reduced steady-state levels of mRNAs (but not of mt-tRNAs or mt-rRNAs), and determining reduced polyadenylation and transcript-processing defects, and impaired translation [149,150,151,152,153]. LRPPRC is also able to block the action of PNPase that degrades RNA, while promoting polyadenylation by stimulating the activity of mtPAP [143]. The other protein that acts in a complex with LRPPRC to stabilize mRNAs is the stem-loop-interacting RNA binding protein (SLIRP) [154]. The knock-down of one of the two proteins causes a decrease in the levels of the other, moreover SLIRP alone is not able to have an effect on polyadenylation of the transcript [143,152]. It has been suggested that LRPPRC is able to stabilize a pool of translationally inactive mt-mRNAs that are not associated with the ribosome. It has also been suggested that the LRPPRC/SLIRP complex can bind the mRNA preventing the formation of secondary structures, leaving the 3′-end available for polyadenylation. This activity could also be involved in the suppression of PNPase/hSUV3 mt-mRNAs degradation [143,152]. 

LRPPRC mutations are causative of the Leigh syndrome French Canadian (LSFC) type and was identified as one of the first nuclear mitochondrial disease genes [155]. The mutation is associated with a severe autosomal recessive disease characterized by severe COX deficiency, which particularly affects the liver and brain, and to a lesser extent fibroblasts and skeletal muscle [156]. Patients affected suffer from a severe neurological disorder characterized by subacute necrotizing encephalopathy, moderate developmental delay, hypotonia, ataxia, strabismus, opthalmoplegia, optic atrophy, and mild facial dysmorphia [155]. In patient fibroblasts, LPRRPC mutation leads to a decrease in the mRNA levels of COI and COIII transcripts and a reduced content of COXI and III proteins explaining the reduced COX activity of these cells [156,157]. Tissue-specific differences were also described for these patients, suggesting that different expression levels of LPRRPC and SLIRP are responsible for the phenotype [158].

LPRRPC has been documented in various tumors, contributing to the apoptosis resistance of human cancer cells. LRPPRC is abundantly expressed in the side population of lung adenocarcinoma cell lines, where cancer stem cells are enriched. The expression of this protein was verified in different types of tumors such as those associated with lung adenocarcinoma, esophageal squamous cell carcinoma, endometrial adenocarcinoma, lymphoma, and stomach, colon, and mammary cancers. In all these cases, LPRRPC was highly expressed, and the knock-down of the protein in lung adenocarcinoma cells reduced the ability of anti-apoptosis, invasion, and in vitro colony formation of the cells, highlighting the fundamental role of LPRRPC in tumorigenesis, resistance to apoptosis, and invasion of cancer cells [159]. LRPPRC has also been identified as an inhibitor of autophagy and mitophagy via interaction with the mitophagy initiator Parkin. Dysfunctions of LRPPRC are associated with poor prognosis in ovarian cancer patients. Moreover, LRPPRC overexpression was found in prostate adenocarcinomas and gastric cancer. Zou and colleagues investigated a possible connection between autophagy inhibition and LRPPRC involvement in cancer development. It was demonstrated that autophagy stimulates Parkin translocation to trigger the rupture of the outer membrane of mitochondria and binds to LRPPRC. The authors showed that the two proteins interact, helping mitochondria to being engulfed in the autophagosomes for the degradation and demonstrating that LRPPRC functions as a checkpoint protein that prevents mitochondria from autophagy degradation [160].

## 9. Conclusions

The human mitochondrial genome is extremely small compared with the nuclear genome; however, despite its reduced dimensions, transcription and translation of mitochondrial genes are essential for cell well-being. Mitochondrial DNA replication and transcription are spatio-temporally regulated to adapt to the metabolic demand of the cell and are controlled by several factors. In MRGs, both strands of mtDNA are transcribed into two polycistronic mtRNA molecules that are processed to release three different RNA species: tRNA, rRNA, and mRNA. Once formed, these RNAs are further processed to reach their mature form and to generate mitochondrial polypeptides. Finally, damaged mtRNA has to be degraded to avoid the formation of an aberrant transcript. 

In the last decades, several mitochondrial human pathologies have been associated with a malfunctioning mtRNA metabolism. However, although many research groups have contributed to elucidating the molecular details of mitochondrial RNA processing, there are still several open questions that need to be answered.

## Figures and Tables

**Figure 1 ijms-20-02221-f001:**
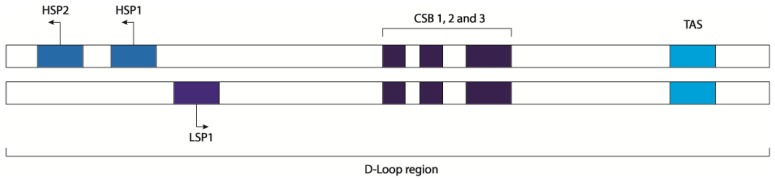
Schematic representation of mitochondrial D-loop region. Heavy strand promoters 1 and 2 (HSP1 and HSP1), light strand promoter 1 (LSP1), conserved sequence blocks 1, 2, and 3 (CSB I, II, and III), and termination-associated sequences (TAS).

**Figure 2 ijms-20-02221-f002:**
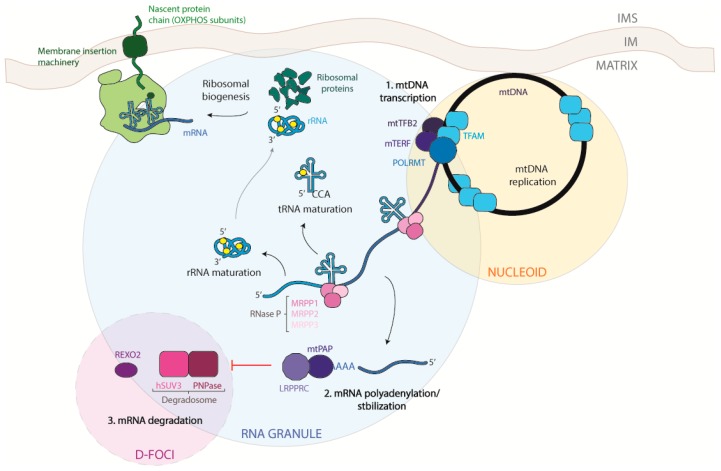
Schematic view of mitochondrial DNA transcription, RNA processing, and degradation. Mitochondrial DNA transcription takes place between nucleoids and mitochondrial RNA granules (MRGs). TFAM is the principal protein needed for the initiation of transcription as long as POLMRT, TFBM1, and TFBM2 (**1**). After transcription of the polycistronic molecules, RNA is immediately processed by RNASE P and Z to release tRNAs, following the tRNA punctuation model. Once tRNAs, mRNAs, and rRNAs are released, the translation process can start (**2**). mRNA degradation takes place in the D-foci, close to the MRGs. PNPase and hSuv3 are components of the degradosome. REXO2 isalso present in this compartment and degrades small RNA oligonucleotides (**3**). Black arrows: transitions; Red arrow: inhibition.

**Table 1 ijms-20-02221-t001:** Factors affecting mitochondrial transcription. The table reports on protein belonging to the three classes of factors (hormones, chromatin remodeling enzymes, and nuclear transcription factors) mentioned in the text for their role in regulating mitochondrial transcription. The protein functions of hormones were obtained from the Human Metabolome Database, and those of all other proteins were obtained from UniProt.

Class	Protein Name	Function	Effect on Mitochondrial Transcription
***Hormones***	**Thyroid hormone T3/receptor**	T3 thyroid hormone is normally synthesized and secreted by the thyroid gland. T3 is a triiodothyronine and is primarily responsible for regulation of metabolism.	Promotes mtDNA transcription directly binding to mtDNA in the D-loop region and in the 12S gene [29,30,31,32].
	**Glucocorticoid hormones/receptor**	Glucocorticosteroids are generally required for stress response and are involved in several processes such as inflammation, allergy, collagen diseases, asthma, adrenocortical deficiency, shock, and some neoplastic conditions.	Promotes mtDNA transcription directly binding to the GR inserted into the inner membrane or to glucocorticoid responsive elements in the mtDNA [34].
	**17β-Estradiol and ERβ**	Estradiol is the most potent form of mammalian estrogenic steroids estradiol is a potent endogenous antioxidant, attenuates induction of redox sensitive transcription factors, hepatocyte apoptosis and hepatic stellate cells activation. It has been reported to induce the production of interferon (INF)-gamma in lymphocytes, and augments an antigen-specific primary antibody response in human peripheral blood mononuclear cells.	Promotes mtDNA transcription; in particular, it increases Complex V gene expression [35].
	**Melatonin**	Melatonin is a biogenic amine produced by the pineal gland. Melatonin regulates the sleep–wake cycle by chemically causing drowsiness and lowering the body temperature. It is also implicated in the regulation of mood, learning and memory, immune activity, dreaming, fertility, and reproduction and is also an effective antioxidant. Most of the actions of melatonin are mediated through the binding and activation of melatonin receptors.	Reduces mtRNA transcription indirectly by influencing the mRNA and protein levels of TFAM and TFB1M and 2M [36].
***Chromatin remodeling enzymes***	**TFAM**	Binds to the mitochondrial light strand promoter and functions in mitochondrial transcription regulation. Required for accurate and efficient promoter recognition by the mitochondrial RNA polymerase. Promotes transcription initiation from the HSP1 and the light strand promoter by binding immediately upstream of transcriptional start sites. Is able to unwind DNA. Bends the mitochondrial light strand promoter DNA into a U-turn shape via its HMG boxes. Required for maintenance of normal levels of mitochondrial DNA. May play a role in organizing and compacting mitochondrial DNA.	Necessary requirement for mtDNA transcription. Methylation of CpG islands can increase TFAM/DNA binding, increasing transcription [52,53].
	**MOF**	Histone acetyltransferase may be involved in transcriptional activation. May influence the function of ATM. It is involved in acetylation of nucleosomal histone H4 producing specifically H4K16ac. It may be involved in acetylation of nucleosomal histone H4 on several lysine residues. It can also acetylate TP53/p53 at ‘Lys-120.’	Promotes mtDNA transcription [57].
	**STAT3**	Signal transducer and transcription activator that mediates cellular responses to interleukins and other growth factors. Acts as a regulator of inflammatory response by regulating differentiation of naive CD4^+^ T-cells into T-helper Th17 or regulatory T-cells. Involved in cell cycle regulation by inducing the expression of key genes for the progression from G1 to S phase.	Negatively influences mtDNA transcription [58].
	**SIRT1**	NAD-dependent protein deacetylase that links transcriptional regulation directly to intracellular energetics and participates in the coordination of several separated cellular functions such as cell cycle, response to DNA damage, metabolism, apoptosis, and autophagy. Can modulate chromatin function through deacetylation of histones and can promote alterations in the methylation of histones and DNA, leading to transcriptional repression.	Negatively influences mtDNA transcription [59].
	**Dnmt**	Methylates CpG residues. Preferentially methylates hemimethylated DNA. Associates with DNA replication sites in S phase maintaining the methylation pattern in the newly synthesized strand. It is responsible for maintaining methylation patterns established in development.	Negatively impacts mtDNA transcription through methylation of the D-loop region.
***Nuclear transcription factors***	**c-Jun**	Transcription factor that recognizes and binds to the enhancer heptamer motif 5′-TGA[CG]TCA-3′. Involved in activated KRAS-mediated transcriptional activation of USP28 in colorectal cancer (CRC) cells.	Decreases mtDNA transcription in concert with retinoid X receptor pathway [38].
	**NFATc1**	Plays a role in the inducible expression of cytokine genes in T-cells, especially in the induction of the IL-2 or IL-4 gene transcription. Also controls gene expression in embryonic cardiac cells. Is required for osteoclastogenesis and regulates many genes important for osteoclast differentiation and function.	Inhibits transcription of Cyt-b and MT-ND1 through binding of the D-loop region [39].
	**NRF1/2**	Transcription factors activate the expression of the EIF2S1 (EIF2-alpha) gene. Links the transcriptional modulation of key metabolic genes to cellular growth and development. Implicated in the control of nuclear genes required for respiration, heme biosynthesis, and mitochondrial DNA transcription and replication.	Fundamental to mtDNA transcription. Promotes expression of TFAM, TFB1M, TFB2M, RNA processing enzymes, PGC1, and PRC [70].
	**PGC1 and PRC**	PGC1 plays the role of a stimulator of transcription factors and nuclear receptors activities. Activates transcriptional activity of estrogen receptor alpha, nuclear respiratory factor 1 (NRF1), and glucocorticoid receptor in the presence of glucocorticoids. PRC acts as a coactivator during transcriptional activation of nuclear genes related to mitochondrial biogenesis and cell growth. It is involved in the transcription co-activation of CREB and NRF1 target genes.	Fundamental to mtDNA transcription. Increases transcription of NRF1 [71].
	**mTOR**	Serine/threonine protein kinase, which is a central regulator of cellular metabolism, growth, and survival in response to hormones, growth factors, nutrients, energy and stress signals. mTOR directly or indirectly regulates the phosphorylation of at least 800 proteins. Functions as part of two structurally and functionally distinct signaling complexes mTORC1 and mTORC2 (mTOR complex 1 and 2).	Increases mtDNA transcription through modulation of PGC and YY1 [73].
	**YY1**	Multifunctional transcription factor that exhibits positive and negative controls on a large number of cellular and viral genes by binding to sites overlapping the transcription start site.	Decreases mtDNA transcription. Needed for the rapamicin-dependent inhibition of mTOR [73].
	**p53**	A multifunctional enzyme that mainly acts as a tumor suppressor in many tumor types and induces growth arrest or apoptosis depending on the physiological circumstances and cell type. Involved in cell cycle regulation as a trans-activator that acts to negatively regulate cell division by controlling a set of genes required for this process.	Fundamental for the maintenance and transcription of mtDNA. Inhibits the entrance of RelA into mitochondria [77].
	**HIF1α**	Functions as a master transcriptional regulator of the adaptive response to hypoxia. Activates, under hypoxic conditions, the transcription of over 40 genes whose protein products increase oxygen delivery or facilitate metabolic adaptation to hypoxia.	Increases mtDNA transcription when overexpressed in mitochondria [78].

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
