# Peer review of "Transcription, Processing, and Decay of Mitochondrial RNA in Health and Disease"

_ijms, 2019, doi:10.3390/ijms20092221_

Round 1
Reviewer 1 Report
1. The manuscript by Barchiesi and Vascotto is a comprehensive review focused on mitochondrial transcription, processing and decay. The authors in the first part of the manuscript provide information on the regulation of transcription by hormones and nuclear transcription factors that have been described to bind mtDNA and direct transcription. They also include a paragraph on mitomiRS which although very limited is useful. However, several additional references should be added (i.e. https://www.ncbi.nlm.nih.gov/pmc/articles/PMC5854179/ and https://www.ncbi.nlm.nih.gov/pubmed/18344181).
2. In addtion to that, a summary of the factors that affect mitochondrial transcription would be most helpful for the readers (either in the form of table or figure).
3. In the second part they summarize current knowledge on processing of transcripts which is covered adequately. Throughout the text, the spelling errors are numerous and should be carefully edited. Overall, the manuscript merits publication, but careful editing, additional references and illustration is required prior to publication.
Author Response
1. The references mentioned by the Reviewer were included and some others have been added in the revised version of the manuscript.
2. Following the Reviewer suggestion, we included a table summarizing the proteins discussed in the text along with their function was added.
3. As suggested by both Reviewers, the text was proofreaded by a professional agency.
Reviewer 2 Report
The review by Barchiesi and Vascotto is very interesting and covers the very relevant topic of mitochondrial RNA expression, processing and turnover and how mutations in the proteins involved in these processes can contribute to human disease. The review itself is rather difficult to read, in part because of English but mainly because it is rather dense, with many protein names mentioned that have to be kept track of by the reader. The latter makes the review very difficult to digest. Below I point out more specific problems and if possible suggestions to address them.
1) While not terrible, the text should be proofread by a professional agency or a native English speaker with a molecular biology background. There are many instances of grammatical mistakes such as “…mtDNA transcription is completely different from the nuclear one and is composed by few proteins:…” (lines 56-57) or “immune‐precipitated” line 272
2) There are a plethora of very long paragraphs that I found myself getting lost in, for example the one encompassing lines 260-288. It would increase the readability to break these long paragraphs into shorter ones organized around more strictly defined topics
3) A table with all proteins discussed, their function, which subchapter(s) they are discussed and human disease caused by their mutation would be very helpful. Perhaps the latter information can be its own table. Perhaps a table of nucleic acid elements would also be helpful, e.g. CSB.
4) Many times, subjects are mentioned before they are properly discussed, which I also found confusing. For example, in lines 63-74, CSB2 is briefly mentioned before the function of CSBs is discussed. I understand that sometimes it is necessary to bring up an element before it is properly discussed, but this should be indicated in the main text as such cases and the occurrence of these events should be reduced as much as possible. The table will also help in this respect.
5) Subchapter 5 is named: “Processing of transcripts” and subchapter 6 is named “Processing of mitochondrial mRNA”. These two sections were hard to read as there is much overlap between them and the data is presented in a somewhat hodgepodge. It would be much easier if the authors spoke about general mechanisms and then highlight differences in mRNA, tRNA and rRNA maturation and degradation pathways. As it is now, even from the titles it is hard to put all this information into proper context.
6) I found subchapter 3.1 not to be particularly convincing and relevant. Many of the cited papers are not definitive, nor is it really relevant for the text as a whole. Wouldn’t it make the text easier to read just to say that many have proposed hormones can control mito gene expression without going into detail of which ones?
7) The discussion of beta-actin lines 245-250 also seems to be expendable. The authors for one do not really say explicitly what beta actin does in the cell. Either this should be expanded or deleted.
8) NADH capping of bacterial RNAs should be cited (Cahova et al, 2014, Nature 519: 374-377.
9) It surprised me that the authors do not mention the mitochondrial tRNAVal or tRNAPhe serving a structural role in mito ribosomes (Brown et al., 2014, Science 346: 718-722; Greber et al., 2014, Nature 515: 283-286).
As far as I know, p53 was sown to be imported into mitochondria under stress conditions for the first time in this paper that should be cited: Vaseva et al., 2012, Cell 149:1536-48.
Author Response
1. As suggested by both Reviewers, the text was proofreaded by a professional agency.
2. We revised the paragraph division to make the text more clear and readable.
3. Following the Reviewer suggestion, we included a table summarizing the proteins discussed in the text along with their function was added.
4. In the revised version we paid attention to avoid mentioning a subject that is described in details later. Unfortunately, in some case that has not been possible.
5. To clarify the point raised by the Reviewer, in the revised version of the manuscript the title of subchapter 5 was changed into “Processing of mitochondrial transcript”, while the subchapter 6 was changed into “Maturation of mitochondrial mRNA”. While in the first one are described the mechanisms and factors involved in the processing of the two polycistronic transcripts, the second one is specifically focused on the post-transcriptional maturation processing of mRNAs.
6. Considering that some readers could find of interest the information regarding the role of hormones in regulating mitochondrial transcription and that Reviewer 1 considered this part relevant, we decided to do not modify the text.
7. In the revised version of the manuscript, the paragraph regarding the beta-acting was removed.
8. As suggested, the reference has been added.
9. In the review we mainly discussed mt-mRNA processing, maturation and degradation. Mitochondrial tRNAs and rRNAs are not a specific focus of this work.
10. Reference on p53 has been updated.